# Influence of Sex on Urinary Organic Acids: A Cross-Sectional Study in Children

**DOI:** 10.3390/ijms21020582

**Published:** 2020-01-16

**Authors:** Marianna Caterino, Margherita Ruoppolo, Guglielmo Rosario Domenico Villani, Emanuela Marchese, Michele Costanzo, Giovanni Sotgiu, Simone Dore, Flavia Franconi, Ilaria Campesi

**Affiliations:** 1Department of Molecular Medicine and Medical Biotechnology, University of Naples ‘Federico II’, 80131 Napoli, Italy; marianna.caterino@unina.it (M.C.); guglielmorosariodomeni.villani@unina.it (G.R.D.V.); michele.costanzo@unina.it (M.C.); 2CEINGE—Biotecnologie Avanzate Scarl, 80145 Naples, Italy; marchese@ceinge.unina.it; 3Department of Mental and Physical Health, Preventive Medicine, University of Campania “Luigi Vanvitelli”, 80138 Naples, Italy; 4Clinical Epidemiology and Medical Statistics Unit, Department of Medical, Surgical and Experimental Sciences, University of Sassari, 07100 Sassari, Italy; gsotgiu@uniss.it (G.S.); simonedore@hotmail.it (S.D.); 5Laboratory of Sex-Gender Medicine, National Institute of Biostructures and Biosystems, 07100 Sassari, Italy; franconi.flavia@gmail.com; 6Department of Biomedical Sciences, University of Sassari, 07100 Sassari, Italy

**Keywords:** urine metabolome, GC, mass spectrometry, sex-specific reference values

## Abstract

The characterization of urinary metabolome, which provides a fingerprint for each individual, is an important step to reach personalized medicine. It is influenced by exogenous and endogenous factors; among them, we investigated sex influences on 72 organic acids measured through GC-MS analysis in the urine of 291 children (152 males; 139 females) aging 1–36 months and stratified in four groups of age. Among the 72 urinary metabolites, in all age groups, 4-hydroxy-butirate and homogentisate are found only in males, whereas 3-hydroxy-dodecanoate, methylcitrate, and phenylacetate are found only in females. Sex differences are still present after age stratification being more numerous during the first 6 months of life. The most relevant sex differences involve the mitochondria homeostasis. In females, citrate cycle, glyoxylate and dicarboxylate metabolism, alanine, aspartate, glutamate, and butanoate metabolism had the highest impact. In males, urinary organic acids were involved in phenylalanine metabolism, citrate cycle, alanine, aspartate and glutamate metabolism, butanoate metabolism, and glyoxylate and dicarboxylate metabolism. In addition, age specifically affected metabolic pathways, the phenylalanine metabolism pathway being affected by age only in males. Relevantly, the age-influenced ranking of metabolic pathways varied in the two sexes. In conclusion, sex deeply influences both quantitatively and qualitatively urinary organic acids levels, the effect of sex being age dependent. Importantly, the sex effects depend on the single organic acid; thus, in some cases the urinary organic acid reference values should be stratified according the sex and age.

## 1. Introduction

The metabolome reflects the real-time dynamics of the reactions and changes that characterize the biochemistry of an organism, contributing to define its phenotype. Metabolomics is a powerful tool to detect changes in different conditions, i.e., health versus disease status, so can be helpful to improve the modern approach of personalized medicine [1,2].

The urine metabolites (e.g., organic acids) derive from food, beverages, drugs, and bacterial by-products, and endogenous processes. Endogenous metabolites can be helpful to assess metabolic pathways and, then, health status, nutritional deficits, and pharmacological responses [3,4,5]. For example, changes in fumarate and malate urinary concentrations are associated with mitochondrial diseases [6]. Among endogenous factors, age and sex seem to have the strongest impact [7,8,9,10,11,12,13,14,15].

The influence of age on few urinary metabolites has been described in children [16,17,18], and in adults [13,19,20]. For example, creatinine increases with age, while creatine, glycine, citrate, succinate, and acetone show a negative correlation with age in 12 year old children [18], while lactate, citrate succinate, N-acetylglutamic acid, dimethylamine, glutamine, 1-methylnicotinamide decrease with age in children from 6 months to 4 years of age [17]. Urine adult metabolome seems to be influenced also by sex [14,19,20,21]. In the urine of adult women, fumarate, succinate, 2-hydroxy-glutarate, malate, Hippurate, and citrate are higher [14,22,23,24], whereas alpha-ketoglutarate, stearate, and 4-hydroxy-butyrate are higher in healthy men [14].

Urinary organic acids usually refer to a broad class of molecules used in physiological processes, that can derived from endogenous metabolism and from dietary protein, fat, and carbohydrate [25]. Urinary organic acids are often adopted to diagnose inborn errors of metabolic disorders; therefore, the identification of distinct reference values for males and females is a key issue [26]. Normal reference values of urinary metabolites have been published for many metabolites without considering the sex of individuals with some exception, as in the case of creatinine [27]. Individual reference values for females and males as a function of their age are importantly needed for the improvement of our understanding of sex- and age-related differences. The above considerations underline the importance of having standardized entry criteria either for appropriateness in diagnosis, therapy and for clinical studies [28]. In fact, FDA guidelines promote collection and analysis of sex-specific data in clinical trials [29].

The aim of our cross-sectional study is to evaluate the role of sex and age on urinary organic acids, in order to establish, if the case, a sex specific reference value, as in the case of creatinine and to study the inter-sectoriality with age.

## 2. Results

A total of 72 urinary metabolites were detected in 152 male and 139 female samples. 4-hydroxy-butirate and homogentisate were found only in males, whereas 3-hydroxy-dodecanoate, methylcitrate, and phenylacetate were found only in females; 19 compounds were found in more than 90% of the analyzed samples, whereas 21 metabolites were detected in 20–90% of the available samples, and 32 metabolites in <20% (Appendix A). Relevantly, 3,4-dihydroxy-butyrate and 5-hydroxy-caproate were found in more than 20% of the recruited females and in less than 20% of male samples (33.8% and 26.6% for 3,4-dihydroxy-butyrate; 0.6% and 17.7% for 5-hydroxy-caproate, respectively).

### 2.1. Intra-Sex Analysis in Male Cohort of Different Ages

In the male cohort, age affected four metabolites: fumarate, succinate, hippurate, and suberate. In detail, fumarate was significantly higher in 1–6 months than in 25–36 months aged males (Figure 1A). Hippurate was significantly higher in age groups 3 and 4 (Figure 1B). In males 1–6 months suberate levels were statistically different if compared with levels found in age group 3 and 4 (Figure 1C), whereas succinate was significantly higher in age groups 4 than in groups 1 and 2 (Figure 1D). Furthermore, four compounds, 2-ethyl-3-hydroxy-propionate, oxalate, tiglylglycine, uracil, were significantly different (Appendix A). Finally, the remaining compounds did not show any significant age-related changes (Appendix A).

### 2.2. Intra-Sex Analysis in Female Cohort of Different Ages

In the female cohort, the age affected 18 (25%) metabolites; 3-hydroxy-isobutyrate and 3-hydroxy-propionate increased with age, being significantly higher in females aged 25–36 months old (Figure 2). The other 16 organic acids showed a fluctuating trend: 2-hydroxy-glutarate, 3-methylglutaconate, citrate, and pyruvate increased from the age group 1 to the age group 3 and, then, decreased in the age group 4. Whereas, alpha-ketoglutarate, azelate, cis-aconitate, ethylmalonate, lactate, and pimelate increased from the age group 1 to the age group 2 and decreased from the age group 2 to the age group 4 (Figure 2). 2-ethyl-3-hydroxy-propionate, 4-hydroxy-phenylacetate, fumarate, glutarate, pyroglutamate, and sebacate, showed a spike only in a specific age group (Figure 2). In particular, 2-ethyl-3-hydroxy-propionate was significantly different when age groups 2 and 3 were compared; 4-hydroxy-phenylacetate and fumarate significantly differed between age groups 1 and 2 (Appendix A). Glutarate was significantly lower in age group 1 if compared with age group 3 (Figure 2), whereas pyroglutamate was significantly lower in age group 1 if compared with age groups 2 and 3. Sebacate was significantly higher in the age group 2 if compared with age groups 1 and 3 (Figure 2). 3-hydroxy-isovalerate and 3-hydroxy-sebacate, were significantly different (Appendix A).

### 2.3. Inter-Sex Analysis

Several compounds displayed sex differences and similarities. In the first 6 months of life, sex differences were more frequent and the majority of urinary organic acids were higher in males (Figure 3A,B), with the exception of 2-ethyl-3-hydroxy-propionate, 2-hydroxy-isobutyrate, 2-methyl-3-hydroxy-butyrate, 3-methyl-glutarate, hippurate, methylsuccinate, stearate, and uracil (Figure 3C,D). Appendix A shows urinary organic acids whose concentrations were similar between males and females. The fold change, calculated as base 2 logarithm of female/male ratio for each significantly divergent metabolite, revealed that the major differences were observed for 2-ethyl-3-hydroxy-propionate, 2-hydroxy-isobutyrate, 2-methyl-3-hydroxy-butyrate, 3-methyl-glutarate, methylsuccinate, and stearate (Figure 4A).

In the second group of age, sex differences decreased, and the majority of metabolites were significantly higher in females, namely 2-ethyl-3-hydroxy-propionate, 2-methyl-3-hydroxy-butyrate, alpha-ketoglutarate, fumarate, methlysuccinate, pimelate, pyruvate, and stearate (Figure 5A). Fold changes showed that 2-ethyl-3-hydroxy-propionate, 2-methyl-3-hydroxy-butyrate, and methylsuccinate were more different between males and females (Figure 4B).

Similarly, in the age group 3, 3-methyl-glutaconate, alpha-ketoglutarate, citrate, glycolate, and oxalate were higher in females, and 2-ethyl-3-hydroxy-propionate, adipate, and hippurate were higher in males (Figure 5B). The highest fold changes were reported for 3-methyl-glutaconate and citrate, followed by alpha-ketoglutarate (Figure 4C).

In the age group 4, 3-hydroxy-propionate, 3-methyl-glutaconate, alpha-ketoglutarate, azelate, and pyruvate showed a significant difference between males and females. All urinary organic acids were found higher in females, with the exception of azelate (Figure 5C), which also showed the highest fold change (Figure 4D).

### 2.4. Metabolomic Intra-Sex Analysis in Male and Female Subjects of Different Ages

The enrichment analysis tool, found that the top-ranking hits resulted in phenylalanine metabolism (*p*-value = 0.0001, impact = 0.03), citrate cycle (*p*-value = 0.001, impact = 0.03), alanine, aspartate and glutamate metabolism (*p*-value = 0.0014, impact = 0.0028), butanoate metabolism (*p*-value = 0.0039, impact = 0.035), and glyoxylate and dicarboxylate metabolism (*p*-value = 0.006, impact = 0.128) in male urinary metabolome (Figure 6A, Appendix A). Metabolites significantly involved in each metabolic pathway are reported in Appendix A.

In females, urinary organic acids were involved in four metabolic pathways: citrate cycle (*p*-value < 0.0001, impact = 0.31), glyoxylate and dicarboxylate metabolism (*p*-value = 0.0001, impact = 0.06), alanine, aspartate and glutamate metabolism (*p*-value = 0.0002, impact = 0.028), butanoate metabolism (*p*-value = 0.001, impact = 0.10). The latter was the most statistically significant and impacted pathway (Figure 6B, Appendix A).

In males and in females, age affected urinary organic acids involved in citrate cycle, alanine, aspartate and glutamate metabolism, in glyoxylate and dicarboxylate metabolism, and butanoate metabolism, whereas phenylalanine metabolism pathway was affected by age only in males. Relevantly, the ranking of metabolic pathways influenced by age varied in the two sexes (Figure 6, Appendix A).

### 2.5. Metabolomic Inter-Sex Analysis

Metabolic pathway analysis plots were also developed using metabolites which displayed a significant difference between males and females in each class of age. In the age group 1 (Figure 7A, Appendix A), the top-ranking impacted canonical pathways were phenylalanine metabolism (*p*-value = 0.0019, impact = 0.03), citrate cycle (*p*-value = 0.0056; impact value = 0.03), alanine, aspartate and glutamate metabolism (*p*-value = 0.0081; impact value = 0.003), and propanoate metabolism (*p*-value = 0.0086; impact value = 0.05).

In the age group 2, (Figure 7B, Appendix A) the top-ranking impacted canonical pathways were citrate cycle (*p*-value < 0.0001; impact value = 0.19), alanine, aspartate and glutamate metabolism (*p*-value < 0.0001; impact value = 0.0028), butanoate metabolism (*p*-value = 0.0001; impact value = 0.10), vitamin B6 metabolism (*p*-value = 0.0025; impact value = 0.038).

On the other hand, glyoxylate and dicarboxylate metabolism (*p*-value < 0.0001; impact value = 0.14), citrate cycle (*p*-value = 0.001; impact value = 0.14), d-glutamine and d-glutamate metabolism (*p*-value = 0.027; impact value = 0.0) were the most significant metabolic pathways in the age group 3 (Figure 7C, Appendix A).

Citrate cycle (*p*-value = 0.0004; impact value = 0.17) alanine, aspartate and glutamate metabolism (*p*-value = 0.0006; impact value = 0.0), vitamin B6 metabolism (*p*-value = 0.0010; impact value = 0.038) and butanoate metabolism (*p*-value = 0.0016; impact value = 0.085) were the top-ranking impacted canonical pathways in the age group 4 (Figure 7D, Appendix A).

Thus, citrate cycle showed significant sex differences in all age groups, whereas alanine, aspartate and glutamate metabolism was sexually different in all groups, with the exception of the age group 3. In addition, sex differences were detected in Vitamin B6 metabolism and butanoate metabolism for the age groups 2 and 4. Phenylalanine and propanoate metabolisms were different between males and females aged 1–6 months, whereas d-glutamine and d-glutamate metabolism showed a sex difference only in the age group 3.

## 3. Discussion

Metabolites in the urine can be a fingerprint for human beings. Our untargeted metabolomics analysis of the urine allowed the identification of several compounds. Sex deeply influences urinary organic acids levels both quantitatively and qualitatively and the sex-induced variations depend on age. In fact, sex differences are mainly observed in the age group 1, where urinary organic acids are globally higher in males. Starting from the 7th month of life, the trend is reversed, and the concentrations of the metabolites become significantly higher in females. In the first 3–6 months of life infants are subjected to mini puberty: male and females infants show relatively high levels of testosterone, estrogens, serum luteinizing hormone, and follicle-stimulating hormone that peaks at different ages [30]. The endocrine system is different in males and females and this could explain, at least partially, why urinary organic acids show more sex-related differences in the first group of age. In line with previous results [16,31,32], sex differences decrease with age but the single age group presents some specificity. It is also evidenced that age effect is greater in males than in females. Consensus was not found on sex differences in infants because Diaz et al. [33] reported sex-differences in the metabolome of newborns aged 1–2 days, whereas Scalabre et al. [34] did not find any statistically significant differences. However, Scalabre and coworkers measured less and different metabolites than we do.

Another key point of the study is the exclusive dependence of qualitative differences by sex. In particular, 4-hydroxy-butyrate and homogentisate are detected only in males suggesting that glutamine metabolism and catabolism of tyrosine and phenylalanine may diverge between sexes [35,36,37]. Sex differences in the tyrosine and phenylalanine metabolism are, at least in part, confirmed from the presence of phenylacetate only in the urine of females. Indeed, phenylacetate is also produced from 2-phenylethylamine, an “endogenous amphetamine” related to central adrenergic functions [38]. In addition, beyond phenylacetate, 3-hydroxy-dodecanoate, and methylcitrate is also detected only in females. Considering that the medium-chain fatty acid 3-hydroxy-dodecanoate is a key sensor of mitochondrial oxidative metabolism [39], the data suggest that mitochondrial metabolism presents sexual dimorphism. The accumulation of methylcitrate, a downstream metabolite of propionate metabolism and the increased levels of 3-hydroxy-propionate suggest that the degradation of branched chain amino acids may differ in male and female children. Previously, it was found that plasma valine was higher in plasma of female neonates [10]. Other minor qualitative differences regard 3,4-dihydroxy-butyrate and 5-hydroxy-caproate that are detected in more than 20% of female samples but in less than 20% of male ones.

Quantitative sex differences predominate in qualitative differences. Specifically, urinary citrate levels increase with age only in females. According to this, boys excrete less citrate than girls, at the beginning of puberty, suggesting that citrate excretion could be influenced by sexual hormones [40]. The importance of sexual hormones is also confirmed by data obtained in pre and postmenopausal women [41,42,43].

Furthermore, sex differences involve metabolites that are intermediates of Krebs cycle [44] such as citrate, cis-aconitate, alpha-ketoglutarate, succinate, fumarate, malate, and oxalate. The involvement of mitochondria is not surprising because in the past it was proven that mitochondria have different sex features [45,46,47]. The differential TCA intermediates could be linked to an imbalance between cell differentiation versus proliferation, trigged by matrix water recycling and deuterium depletion, via hydratase reactions [48,49]. Indeed, hydrogen/deuterium ratio in cells is considered responsible for growth signaling regulations [50].

The central role of mitochondria in sex-related differences of urinary organic acid is also confirmed by the variation of one of the most impacted pathways, the butanoate metabolism, which is linked to Krebs cycle intermediates, glycolysis, and glutamate synthesis [51]. Further, butanoate can generate ketone bodies, and short-chain lipids [51].

Undigested food is fermented by the gut microbiota producing various microbial metabolites such as short-chain fatty acids such as acetate, propionate, azelate, while others may be co-produced by bacteria and by the host such as butyrate and hippurate [52,53,54,55]. In particular, intestinal micro-organisms produce benzoate [56,57], that the host conjugates with glycine through glycine N-acyltransferase [58], whose gene is less expressed in women than in men [59]. Thus, it is plausible that some differences in urinary organic acids could depend on gut microbiota, which is sex and age dependent [60,61,62], in other words, it is conceivable that gut microbiota may be a generator of sex differences.

The dicarboxylic acids suberate, azelate, pimelate are present in modest amounts in the urine whereas adipate are found in moderate amount. The four dicarboxylic acids are differently affected by age and by sex. Suberate, a product of ω-oxidation [63,64], decreases with age only in males, while azelate, a product of colonic bacteria, and pimelate are age dependent only in females. Their urinary levels increase in disorders of mitochondrial β-oxidation and peroxisomal β-oxidation, for which they are very significant for diagnostic purposes [32].

The end product of glycolysis, pyruvate, is involved in all examined metabolic pathways (citrate cycle, glyoxylate and dicarboxylate metabolism, alanine, aspartate and glutamate metabolism, and butanoate metabolism). Under anaerobic conditions, pyruvate is converted by lactate dehydrogenase to lactic acid. Lactic acid is a classic marker of mitochondrial metabolic dysfunction together with acylcarnitines [65]. Discrete amounts of pyruvate and lactate are present in the urine of both sexes presenting sex quantitative differences. Previously it has been shown that blood acylcarnitines are overall higher in male neonates than in female ones [10]. The higher urinary levels of lactate and higher levels of acetylcarnitines suggest that male mitochondria of neonates function less in comparison to female ones.

Globally, our results provide some evidence suggesting that sex impacts qualitatively and quantitatively urinary metabolome in male and female infants and children. In view of observed differences, our results suggest that baseline sex-related differences should be considered in future preclinical and clinical studies. In the past and also in the present [28,66], sex has not been adequately considered, but actually there is mounting evidence of its importance in diagnoses and therapies because sex and gender influence pharmacological responses [67,68,69].

In addition, the influence of sex is linked to age. Thus, the sex stratification in every single class of age is mandatory in the validation of urinary organic acid reference values. Knowledge of the reference values for urinary organic acids in a healthy pediatric population is important, applying reference values specific for a single sex to another sex being inappropriate. Then, the role of sex should be taken in account for a personalized diagnosis, and to give more rigorous scientific background with better standardized entry criteria in studies on biomarkers as already suggested [9,10,14,28,70,71,72].

These data also indicate that sex differences start early in life. It is also important and relevant to highlight the link between sex and age. This is a clear example of intersectoriality: multisectorial and intersectorial aspects are, in fact, crucial for health and well-being as also declared by the World Health Organization [73].

## 4. Methods

### 4.1. Populations

A cross-sectional study was carried out during the period 2009–2014 to screen metabolism inborn error (research project performed in the Campania region, Italy). A total of 291 urinary samples, 152 from males and 139 from females, were collected from healthy Caucasian children (aged 1 to 36 months). Donors were divided into four age groups: from 1 to 6 months (age group 1: 37 males and 37 females), from 7 to 12 months (age group 2: 30 males and 28 females), from 13 to 24 months (age group 3: 43 males and 38 females), and from 25 to 36 months (age group 4: 42 males and 36 females). Informed consent was obtained from subjects’ parents, and protocols were reviewed and experimental protocols followed guidelines approved by Italian Ministry of Health in the law 167 of 19 August 2016).

### 4.2. Urine Collection

Urine samples were collected by using a special plastic bag with a sticky strip on one end, made to fit over genital area; urine was, then, recovered by medical operator from the bag to a test tube. An aliquot of each urine sample was immediately centrifuged at 2000 rpm for 10 min and urine creatinine concentration was measured using standard procedures. The other aliquots were frozen in dry ice and stored at −80 °C.

The urine creatinine was measured in triplicate by colorimetric method according to Jaffè reaction [74].

### 4.3. Preparation of Samples for GC-MS Analysis of Metabolites

A urine volume containing 0.5 mmol of creatinine was analyzed, in order to normalize metabolites differences affected by hydratation. Urine pH was adjusted to 14 by adding 30% NaOH.

Urine metabolites were oxidized with 0.5 mL of hydroxylamine hydrochloride (2.5 mg/mL in water) and heated at 60 °C for 60 min. After cooling, the samples were acidified by using several drops of 2.5 M H_2_SO_4_ until pH 1 was reached. Internal standards, dimethylmalonic acid (10 mg/mL in 1:1 of H_2_O/CH_3_CH_2_OH (*v*:*v*)), pentadecanoic acid (10 mg/mL in CH_3_CH_2_OH), and tropic acid (10 mg/mL in H_2_O) (Sigma-Aldrich, St. Louis, MO, USA) were prepared and then added to the sample where they had the final concentrations of 10 μM. Samples were extracted three times with 2 mL of ethyl acetate and centrifuged at 3000 rpm for 5 min. At the end of centrifugation, the organic phase was collected and saturated with approximately 1 g of Na_2_SO_4_ (Sigma-Aldrich, St. Louis, MO, USA) for 60 min at RT to remove water. After centrifugation (3000 rpm for 10 min), the organic phase was collected and evaporated at 40 °C under a gentle nitrogen flow.

Finally, the sample was derivatized in 50 µL of *N*,*O*-bis(trimethylsilyl)trifluoroacetamide (Sigma-Aldrich, St. Louis, MO, USA) at 60 °C for 30 min. The mixture was cooled at RT and transferred into 250 µL conical glass inserts in order to carry out the subsequent GC-MS analysis.

### 4.4. GC-MS Analysis

GC-MS analysis was performed using an Agilent 7890A (Agilent Technologies, Santa Clara, CA, USA) gas chromatograph coupled with an Agilent 5975C (Agilent Technologies, Santa Clara, CA, USA) mass spectrometer on GC column HP-5MS; 30 m × 0.250 mm × 0.25 µm. The GC was equipped with a split-mode capillary injection port held at 280 °C with a split ratio of 10:1. The chromatographic gradient was programmed from 70 to 280 °C at a rate of 10 °C/min with a helium flow of 1 mL/min. The mass spectrometer operated in scan mode into a scan range from 50 to 650 amu, using a 6 min solvent delay. The metabolites were identified by the NIST and Wiley mass spectra library (release 2008), and the MSD Productivity Chemstation software (Agilent Technologies, Santa Clara, CA, USA). For each compound the typical and unique fragmentation pattern, the mass charge ratios and each peak abundance were compared with the mass spectra, present in the NIST spectra library using ChemStation Software. A list of compound similarities was obtained from each mass spectra: peaks with similarity index more than 80% were given a compound name, while those having less than 80% similarity were listed as unknown metabolites. Compounds were quantified using the areas of total ion chromatograms, comparing the area of each compound with the area of the internal standards used at known concentrations. Concentrations of metabolites were expressed as mmol metabolite/mol creatinine. The internal quality controls were represented by the areas of internal standard added to the initial mixture [75].

### 4.5. Pathway Analysis

The interpretation of the metabolomic dataset was carried out using a free metabolomics platform, MetaboAnalyst 4.0 (http://www.metaboanalyst.ca). Human Metabolome Database was used to assign the correct number ID (HMDB ID) to each metabolite. Metabolomic dataset was clustered by Metabolite Set Enrichment Analysis (MSEA) and Metabolic Pathway Analysis (MetPA), setting pathway-associated metabolite sets. MSEA is defined as a way to identify biologically meaningful patterns that are significantly enriched in quantitative metabolomic data. MetPA summarizes the most significant metabolic pathways, according to their *p*-value from Enrichment Analysis and impact value. It has the potential to identify subtle but consistent changes in a group of related compounds, which may remain undetected with the conventional approaches.

Each pathway was classified according to matched number from the uploaded data (match status), statistical *p*-values (*p*), *p*-value adjusted by Holm–Bonferroni method (Holm p), *p*-value adjusted using False Discovery Rate (FDR p), pathway impact value calculated from pathway topology analysis (Impact).

### 4.6. Statistical Analysis

Shapiro–Wilk normality test was performed to evaluate the normal distribution of continuous variables. The urinary organic acids detected in less than 20% of the samples were excluded from the analysis because the sample size did not allow to perform a powered statistical analysis and are shown in Appendix A. Median values of the above specified variables for the male and female cohorts were reported. Quantitative variables were described with medians and inter-quartile ranges (IQR) for their non-parametric distribution. Statistical differences between urinary organic acid concentrations were assessed using the Mann–Withney U test and the Kruskall–Wallis test to compare 2 and >2 groups, respectively. Multiple comparisons were performed using Dunn’s test. Differences were considered statistically significant when two-tailed *p*-values were <0.05.

Analyses were performed with Stata version 13 statistical software (StataCorp, College Station, TX, USA).

## Figures and Tables

**Figure 1 ijms-21-00582-f001:**
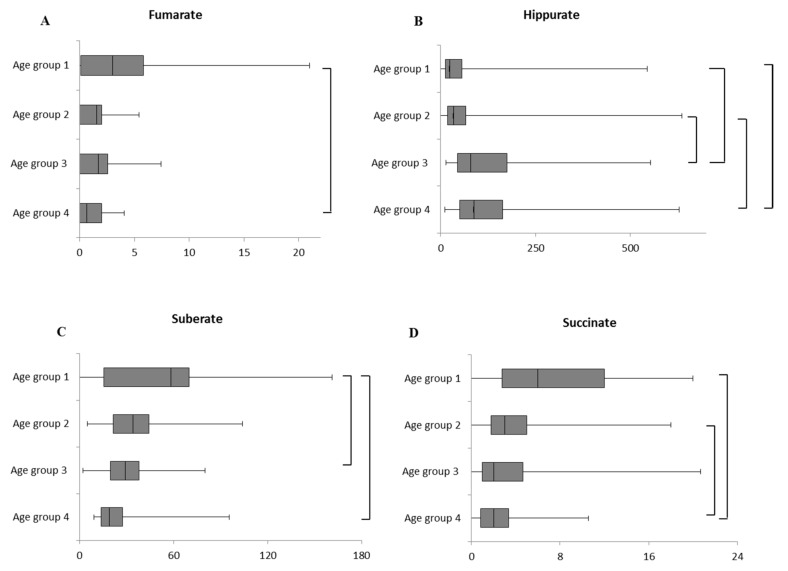
Box plot of age effect on urinary (**A**) Fumarate, (**B**) Hippurate, (**C**) Suberate and (**D**) Succinate (mmol/mol creatinine) in male cohorts. The vertical line across the box represents the median, and the box comprises the first and the third quartiles. The horizontal lines represent the minimum and the maximum values. Sample size for each age group is reported in Appendix A.

**Figure 2 ijms-21-00582-f002:**
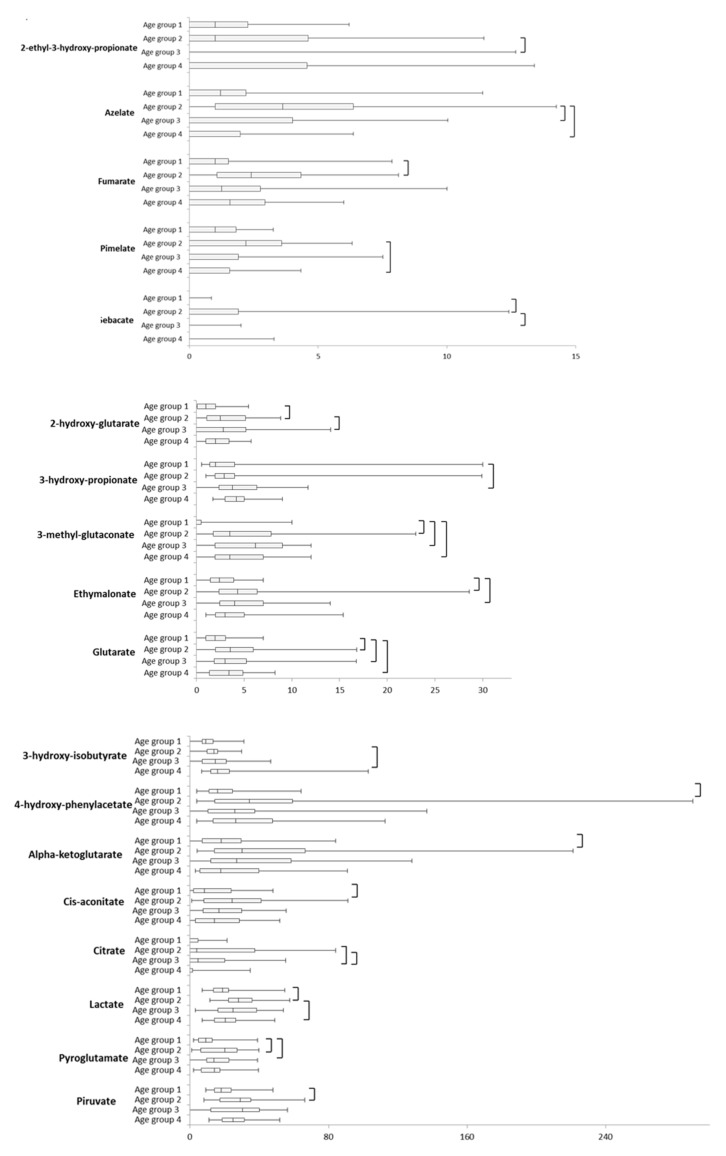
Box plot of age effect on urinary organic acids (mmol/mol creatinine) in female cohorts. The vertical line across the box represents the median, and the box comprises the first and the third quartiles. The horizontal lines represent the minimum and the maximum values. Connectors represent a *p* < 0.05. Sample size for each age group was reported in the method section.

**Figure 3 ijms-21-00582-f003:**
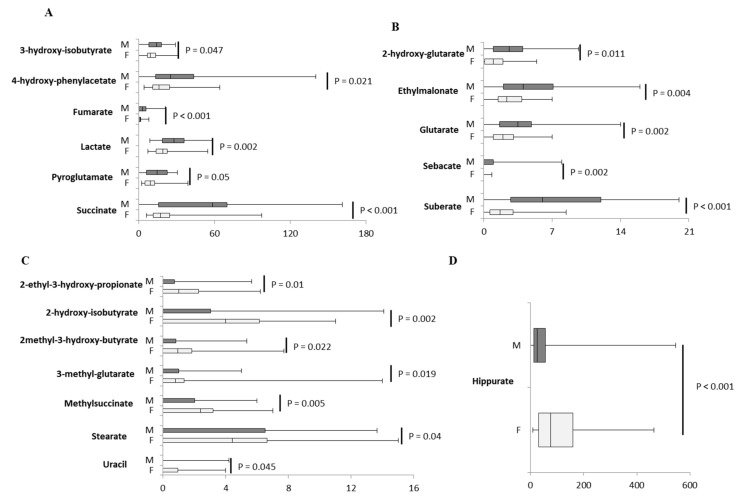
Box plot of urinary organic acids (mmol/mol creatinine) that sexually diverged in age group 1. Panels (**A**–**C**) reported metabolites divided according to the scale of values; (**D**) Sex differenc in hippurate levels. The vertical line across the box represents the median, and the box comprises the first and the third quartiles. The horizontal lines represent the minimum and the maximum values. *p*-values for each urinary organic acid were reported. Sample size for each group is reported in Appendix A.

**Figure 4 ijms-21-00582-f004:**
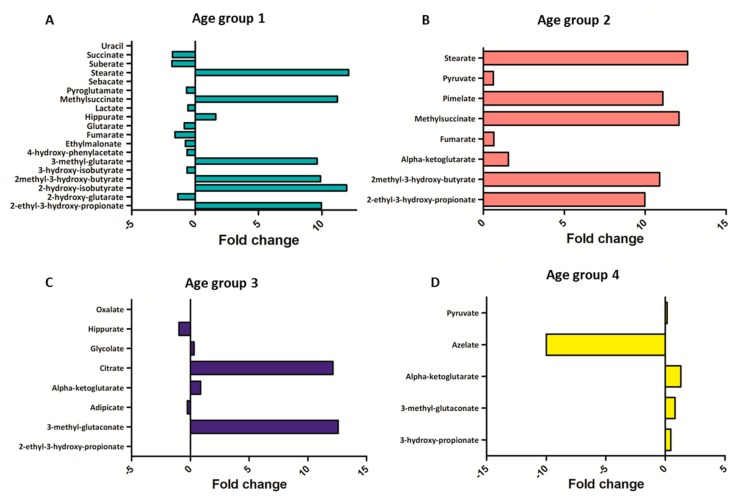
Fold change (log_2_ female/male ratio) calculated for each sexually divergent metabolite in (**A**) age group 1; (**B**) age group 2; (**C**) age group 3; (**D**) age group 4.

**Figure 5 ijms-21-00582-f005:**
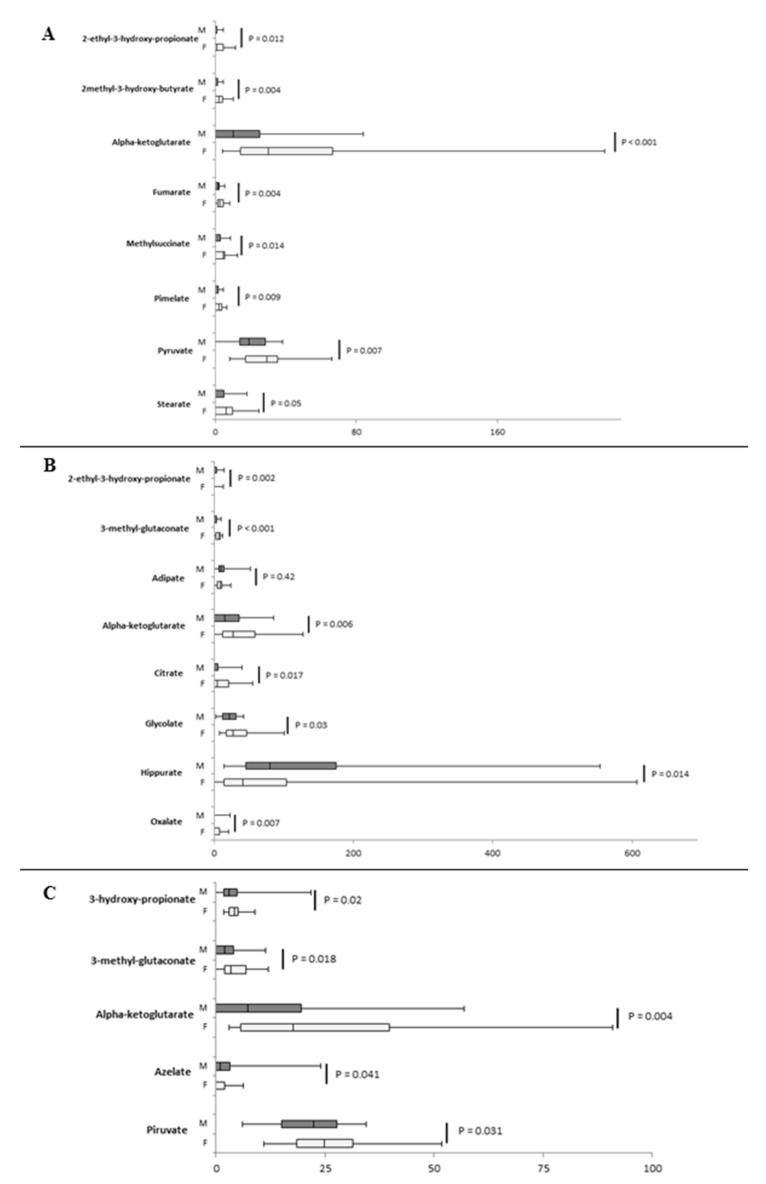
Box plot of urinary organic acids, which displayed a sex difference in age groups 2 (**A**), 3 (**B**), and 4 (**C**). Values are expressed as mmol/mol of creatinine. The vertical line across the box represents the median, and the box comprises the first and the third quartiles. The horizontal lines represent the minimum and the maximum values. Sample size for each group is reported in Appendix A.

**Figure 6 ijms-21-00582-f006:**
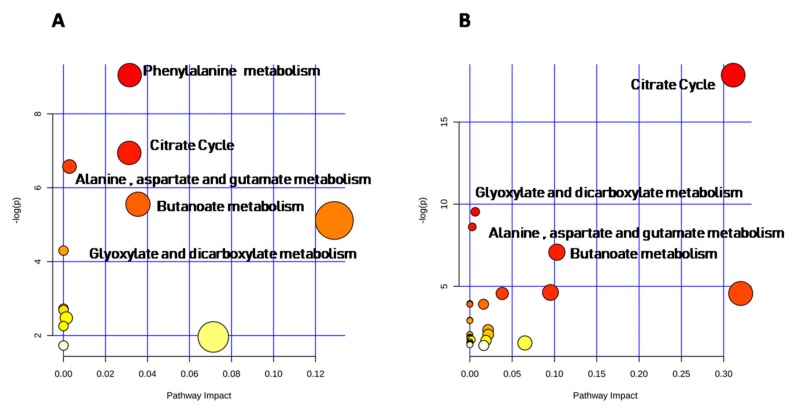
Metabolic pathway analysis for males (**A**) and females (**B**). Pathways analysis nodes were represented as a function of −log (*p*). Nodes color and size were based on *p*-values and the statistical significance (impact value), respectively.

**Figure 7 ijms-21-00582-f007:**
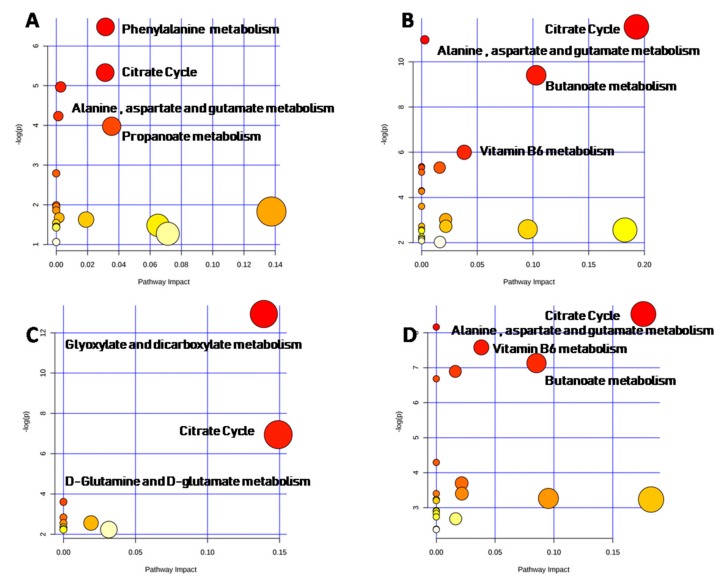
Metabolic pathway analysis for age. Pathway analysis plots were developed as function of the age of the subjects: age group 1(**A**), age group 2 (**B**), age group 3 (**C**), age group 4 (**D**).

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
