# Peer review of "Influence of Sex on Urinary Organic Acids: A Cross-Sectional Study in Children"

_ijms, 2020, doi:10.3390/ijms21020582_

Round 1
Reviewer 1 Report
The manuscript entitled "Sex-dependent urinary metabolome: a cross-sectional study in children" it describes an application of a workflow to an interesting and unique cohort. The reviewer appreciate the large amount of work behind, and thinks it could fit the quality and requirements to be considered for publication in this Journal if some changes are made. I would like to make some suggestions and comments which could help to improve the manuscript.
Title: This title is perhaps too ambitious for this study. It mentions urinary metabolome, and the data doesn't not show it. In urine can be detected and quantified much more different other metabolites and species than the ones presented in this manuscript.
Abstract: At lines 28 to 36 it is repeated twice the same information, first describe in detail females, then males and then both. Just remove one of the sentences to avoid redundant information.
Introduction: It is a too short and general introduction. It does describe the differences in adults and shortly mention children, while the study is focused in child, one expect at least a sentence on how this data relates to yours study. Furthermore I suggest to add few more words discussing why is your gender study important and need it.
Methods: The reference provided for the urine collection does not describe the urine collection. It is important to know in this case if for the younger kids urine was collected from the diapers, which may affect some of the results, like presence of some short fatty acids which can come from the plasticizers and so on.
Please provide a reference for the creatinine measurements, there is more than one standard procedure for measuring creatinine, from ELISA methods to MS methods.
The sample treatment applied in this study is quite aggressive for the sample. One can expect large number of artifacts.
Results: As mentioned before the title suggest a urinary metabolome study, and the results presented show some aminoacids and organic acids altered.
Discussion: How the authors explain some of the large deviations observed?
Lines 273-279, did the authors consider to measure the levels if sex hormones in this study? The reviewer thinks it would add some more value to the manuscript. "Mini puberty occurs"?
Line 288: please rewrite this sentence.
It is suggested in the discussion some of the metabolites are due to the microbiota differences. Gut microbiota can variate a lot from one individual to another, so how the authors can attribute the changes on these metabolites to age and gender?
Tables S4 to S9 provide a list of different pathways altered, but in each significant metabolic pathway, the authors found the same two to three metabolites altered.
Author Response
The manuscript entitled "Sex-dependent urinary metabolome: a cross-sectional study in children" it describes an application of a workflow to an interesting and unique cohort. The reviewer appreciate the large amount of work behind, and thinks it could fit the quality and requirements to be considered for publication in this Journal if some changes are made. I would like to make some suggestions and comments which could help to improve the manuscript.
Title: This title is perhaps too ambitious for this study. It mentions urinary metabolome, and the data doesn't not show it. In urine can be detected and quantified much more different other metabolites and species than the ones presented in this manuscript. REPLY: I thank the referee for his/her suggestion, the title has been modified
Abstract: At lines 28 to 36 it is repeated twice the same information, first describe in detail females, then males and then both. Just remove one of the sentences to avoid redundant information. REPLY:I thank the referee for his/her suggestion, results in the abstract have been better described.
Introduction: It is a too short and general introduction. It does describe the differences in adults and shortly mention children, while the study is focused in child, one expect at least a sentence on how this data relates to yours study. Furthermore I suggest to add few more words discussing why is your gender study important and need it. REPLY:I thank the referee for his/her suggestion, the introduction has been modified
Methods: The reference provided for the urine collection does not describe the urine collection. It is important to know in this case if for the younger kids urine was collected from the diapers, which may affect some of the results, like presence of some short fatty acids which can come from the plasticizers and so on. REPLY:Urine was collected by using special plastic bag with a sticky strip on one end, made to fit over your baby's genital area. The urine in the bag was recovered by medical operator. This information was included in the manuscript
Please provide a reference for the creatinine measurements, there is more than one standard procedure for measuring creatinine, from ELISA methods to MS methods. REPLY:The urine creatinine level was measured by colorimetric method according to Jaffè reaction. (Butler AR. The Jaffé reaction. Identification of the coloured species. Clin Chim Acta. 1975;59(2):227‐232). This information was included in the manuscript
The sample treatment applied in this study is quite aggressive for the sample. One can expect large number of artifacts. REPLY:GC-MS methods have long been used to comprehensively characterize the chemical content of human urine (Bouatra S et al.. PLoS One. 2013 Sep 4;8(9):e73076.) and the sample handling is very well standardized and worldwide used in the laboratories studying inherited metabolic diseases (Villani GR, Gallo G, Scolamiero E, Salvatore F, Ruoppolo M. "Classical organic acidurias": diagnosis and pathogenesis. Clin Exp Med. 2017 Aug;17(3):305-323; la Marca G, Rizzo C. Analysis of organic acids and acylglycines for the diagnosis of related inborn errors of metabolism by GC- and HPLC-MS. Methods Mol Biol. 2011;708:73-98; Phipps WS, Jones PM, Patel K. Amino and organic acid analysis: Essential tools in the diagnosis of inborn errors of metabolism. Adv Clin Chem. 2019;92:59-103). The authors are aware that the global analysis of the urinary metabolome requires the use of different extraction protocols to detect metabolites, characterized by different chemical-physical properties.
Results: As mentioned before the title suggest a urinary metabolome study, and the results presented show some amino acids and organic acids altered. REPLY:I thank the referee for his/her suggestion, the title has been modified
Discussion: How the authors explain some of the large deviations observed?REPLY: If the referee refers to the variability in the samples, this can be attributed, in the first instance, to the inter-individual variability, as single subjects have been analyzed. In addition, the variables analyzed (sex and age) contribute to variability. The key message of the work is, in fact, the need to have specific reference ranges that take into account sex and age, hoping that these aspects are no longer forgotten in clinical research and in daily clinical practice. According to the principles of personalized medicine, it is no longer possible to extrapolate data from a general population, not stratified by sex and age, and adapt it to all.
Lines 273-279, did the authors consider to measure the levels of sex hormones in this study? The reviewer thinks it would add some more value to the manuscript. REPLY:I thank the referee for his/her suggestions. Unfortunately, sex hormones were not measured as sample collection is part of a standard procedure to screen metabolism inborn error of metabolism. "Mini puberty occurs"? the term has been changed with “are subjected to"
Line 288: please rewrite this sentence. REPLY:Rewritten
It is suggested in the discussion some of the metabolites are due to the microbiota differences. Gut microbiota can variate a lot from one individual to another, so how the authors can attribute the changes on these metabolites to age and gender? REPLY:Some differences in urinary organic acids could depend on gut microbiota as some metabolites may also derive in part from the microbiota, which is also dependent on age and sex, thus can participates to the differences. However, this is only a hypothesis therefore we modify the sentence, but further features characterizing this problem are certainly necessary.
Tables S4 to S9 provide a list of different pathways altered, but in each significant metabolic pathway, the authors found the same two to three metabolites altered.REPLY: We only analyzed the pathways in which the metabolites showing significant differences between sexes; therefore the software returns the highest impact pathways in which they are involved. So, different metabolic pathways could present common metabolites.
Reviewer 2 Report
The study of Caterino et al. is potentially a significant contribution to clinical and translational metabolomics after a precise (mechanistic) positioning of the reported urine metabolome findings in the Krebs-Szent-Györgyi (TCA) cycle is considered by the authors. It is because this metabolic cycle provides clinically relevant deuterium (heavy hydrogen) depleting metabolic (matrix) water recycling functions in its cataplerotic (cycling) modus operandi, as described previously (https://doi.org/10.1016/j.mehy.2015.11.016) and applied clinically (https://doi.org/10.1093/neuonc/now284). The paper reports observations that further our understanding of the critically important functions of mitochondria in hydrogen (proton) transfers from nutrients to metabolic water as a function of sex related life expectancy by heavy hydrogen (deuterium) discrimination that determines eukaryote cell differentiation and disease (https://doi.org/10.1016/0014-5793(93)81479-J). The authors excellently lead and cite in the Introduction that urine adult metabolome is influenced by sex as in the urine of adult women, who generally exhibit longer life expectancy; levels of citrate, succinate, fumarate and malate are higher, whereas in healthy men alpha-ketoglutarate, stearate, and 4-hydroxy-butyrate are higher. Medical and translational deutenomics Discussions of the Introduction could indeed confirm that citrate, succinate, fumarate and malate TCA cycle intermediates are for matrix water recycling and deuterium depletion (deupletion) via hydratase reactions that promote cell differentiation versus proliferation, hence women have longer life expectancies than do men, who rather exhibit the branching of the Cycle at alpha-ketoglutarate with an inefficient deuterium depleting ketogenic stearate and hydroxy-butyrate oxidation pattern as indicated by their increase in disposed ketone substrates upon urine analyses in men.
Although the conclusions are correct that “The most relevant sex differences involve the mitochondria homeostasis.”, mitochondrial deupleting functions need further discussion in the age group cohorts as described above for the adult urine findings in a devoted paragraph in Discussions with clinical references discussing the topic already in IJMS already as https://doi.org/10.3390/ijms20204984 and by Nature Springer as https://doi.org/10.1007/s11306-016-0961-5
Author Response
The study of Caterino et al. is potentially a significant contribution to clinical and translational metabolomics after a precise (mechanistic) positioning of the reported urine metabolome findings in the Krebs-Szent-Györgyi (TCA) cycle is considered by the authors. It is because this metabolic cycle provides clinically relevant deuterium (heavy hydrogen) depleting metabolic (matrix) water recycling functions in its cataplerotic (cycling) modus operandi, as described previously (https://doi.org/10.1016/j.mehy.2015.11.016) and applied clinically (https://doi.org/10.1093/neuonc/now284). The paper reports observations that further our understanding of the critically important functions of mitochondria in hydrogen (proton) transfers from nutrients to metabolic water as a function of sex related life expectancy by heavy hydrogen (deuterium) discrimination that determines eukaryote cell differentiation and disease (https://doi.org/10.1016/0014-5793(93)81479-J). The authors excellently lead and cite in the Introduction that urine adult metabolome is influenced by sex as in the urine of adult women, who generally exhibit longer life expectancy; levels of citrate, succinate, fumarate and malate are higher, whereas in healthy men alpha-ketoglutarate, stearate, and 4-hydroxy-butyrate are higher. Medical and translational deutenomics Discussions of the Introduction could indeed confirm that citrate, succinate, fumarate and malate TCA cycle intermediates are for matrix water recycling and deuterium depletion (deupletion) via hydratase reactions that promote cell differentiation versus proliferation, hence women have longer life expectancies than do men, who rather exhibit the branching of the Cycle at alpha-ketoglutarate with an inefficient deuterium depleting ketogenic stearate and hydroxy-butyrate oxidation pattern as indicated by their increase in disposed ketone substrates upon urine analyses in men.
Although the conclusions are correct that “The most relevant sex differences involve the mitochondria homeostasis.”, mitochondrial deupleting functions need further discussion in the age group cohorts as described above for the adult urine findings in a devoted paragraph in Discussions with clinical references discussing the topic already in IJMS already as https://doi.org/10.3390/ijms20204984 and by Nature Springer as https://doi.org/10.1007/s11306-016-0961-5
REPLY: I thank the referee for his/her interesting comment. Some consideration and the suggested references have been added to the discussion. However, I believe that this specific field, deuterium depletion and its role in mitochondria function, goes behind the topic of our observational study. Besides, I consider it an excellent point of reflection which may give further clarification to the issue.
Reviewer 3 Report
Review of manuscript ijms-671291 entitled “Sex-dependent urinary metabolome: a cross-sectional study in children”
SUMMARY
In this manuscript, the authors present a study on the urine metabolome to determine characteristic metabolites (organic acids) for both sex and age factors.
COMMENTARIES
In my opinion, this manuscript is mainly well-written and describes the methodology correctly and results obtained by the authors. However, I think that some major flaws prevent its publication.
* Overall
I feel that the experimental results do not totally support the conclusions reached by the authors. In particular, as the authors found that both age and sex factors were relevant for sample differentiation, I think that both factors should be considered at the same time in a single analysis. In the current version, the interaction between these factors was not considered and, therefore, this could introduce a bias in the obtained results.
* Other issues
- Abstract.
Rewrite three first sentences.
- Introduction.
This section should be greatly expanded. More details and discussion should be included regarding: i) impact of age and sex; ii) effect of age on which urinary metabolites (children and adults); iii) impact of diet.
- Methods.
More details about the sample cohort should be given. How was the distribution of samples in the different age groups? Are there any relevant differences in the diet of the different sex/ages? Many factors could affect the obtained results that, at least, some details should be given about how have been controlled.
How were the metabolites identified? Some details comparing experimental and reference spectra should be given. Nothing is shown in the current version.
How were the QCs prepared? How were they used?
- Results.
In general, all boxplots figures did not provide enough quality information to justify the differences observed in sex/age. Authors should include a visualization able to clearly show the distribution of the concentration values of the considered metabolites. In many cases, it is difficult to know if the large whiskers were caused for a single outlier value or several values (affecting the observed mean).
Also, Figures 2,3 and 4 should be redesigned to facilitate the interpretation of the obtained results.
Author Response
Review of manuscript ijms-671291 entitled “Sex-dependent urinary metabolome: a cross-sectional study in children”
SUMMARY
In this manuscript, the authors present a study on the urine metabolome to determine characteristic metabolites (organic acids) for both sex and age factors.
COMMENTARIES
In my opinion, this manuscript is mainly well-written and describes the methodology correctly and results obtained by the authors. However, I think that some major flaws prevent its publication.
* Overall
I feel that the experimental results do not totally support the conclusions reached by the authors. In particular, as the authors found that both age and sex factors were relevant for sample differentiation, I think that both factors should be considered at the same time in a single analysis. In the current version, the interaction between these factors was not considered and, therefore, this could introduce a bias in the obtained results. REPLY: I thank the referee for his/her comment. However, the initial assumption on the confounding role played by age and sex was statistically proved stratifying the results. In the supplementary materials, tables focused on males and females and different age categories clearly show differences of metabolite concentrations, which depend on sex and age.
* Other issues
- Abstract.
Rewrite three first sentences.: REPLY:rewritten
- Introduction.
This section should be greatly expanded. More details and discussion should be included regarding: i) impact of age and sex; ii) effect of age on which urinary metabolites (children and adults); iii) impact of diet. REPLY: I thank the referee for his/her suggestion, the introduction has been modified
- Methods.
More details about the sample cohort should be given. How was the distribution of samples in the different age groups? I apologise for the oversight. The distribution of samples in the different age group has been now reported in the paragraph 2.1 “populations”: Are there any relevant differences in the diet of the different sex/ages? Many factors could affect the obtained results that, at least, some details should be given about how have been controlled. REPLY:I thank the referee for his/her suggestions. Unfortunately, sample collection is part of a standard procedure to screen metabolism inborn error and information about diet was not available in the study period. In fact, this information computerized access to full patient information is available, in our region, only from 2019.
How were the metabolites identified? Some details comparing experimental and reference spectra should be given. Nothing is shown in the current version. REPLY:The chromatograms were analyzed: each chromatographic peak was check manually, and then used to the area integration. The relative intensity of each peak was normalized against the internal standard in GC/MS run. The metabolite identification was carried out by using NIST (National Institute of Standards and Technology) mass spectra library by the ChemStation Software. Each compound identification was performed comparing the typical and unique fragmentation pattern, the mass charge ratios and each peak abundance versus the mass spectra, present in the NIST spectra library by the ChemStation Software. To each mass spectra comparing a list of compound similarities was obtained. Peaks with similarity index more than 80% were assigned compound names, while those having less than 80% similarity were listed as unknown metabolites (Wu H, Xue R, Dong L, Liu T, Deng C, Zeng H, Shen X. Metabolomic profiling of human urine in hepatocellular carcinoma patients using gas chromatography/mass spectrometry. Anal Chim Acta. 2009 Aug 19;648(1):98-104). This information is now included in the manuscript.
How were the QCs prepared? How were they used? REPLY:Using the term quality control and internal standard in the same sentence we have generated confusion in the reader. In fact, in the methods section we used the term ‘quality controls’ as internal quality controls. They are represented by the following mixture: the stock solutions, 10mg/ml dimethylmalonic acid (MW 132.12) in 1:1 of H2O/CH3CH2OH (v:v), 10mg/ml tropic acid (MW 166.2) in H2O and 10mg/ml pentadecanoic acid (MW MW 242.41) in CH3CH2OH are diluited to 10uM in final mixture, containing sample. In addition, in order to evaluate and assess the laboratories ability to detect inherited disorders resulting in recognizable patterns of organic acid excretion, we used also an external quality control (QC) provided by ERNDIM. Our laboratory analytical capability establishing or excluding specific diagnosis was test by ERNDIM program; ERNDIM is an independent not-for-profit foundation which has been providing External Quality Assurance (EQA) schemes. The external QC are listed below: 2-Methylcitric acid; 4-Hydroxy-Butyric acid; Methylmalonic acid; 2- Hydroxy -Glutaric acid; Adipic acid; Mevalonic acid; 3-Methylglutaconic acid; Creatinine; N-acetylaspartic acid; 3-Methylglutaric acid; Ethylmalonic acid; Pyro glutamic acid; 3- Hydroxy -3-Methylglutaric acid; Fumaric acid Sebacic acid 3- Hydroxy -Butyric acid; Glutaric acid; Suberic acid; 3- Hydroxy -Glutaric acid; Hexanoylglycine; Suberylglycine; 3- Hydroxy -Isovaleric acid; Isovalerylglycine; Tiglylglycine; 3- Hydroxy -Propionic acid; Keto-glutaric acid; Vanillactic acid.
- Results.
In general, all boxplots figures did not provide enough quality information to justify the differences observed in sex/age. Authors should include a visualization able to clearly show the distribution of the concentration values of the considered metabolites. In many cases, it is difficult to know if the large whiskers were caused for a single outlier value or several values (affecting the observed mean).Also, Figures 2,3 and 4 should be redesigned to facilitate the interpretation of the obtained results. REPLY:I thank the referee for his/her suggestion, a clear visualization of the values is shown in the supplementary tables. Moreover, we used figures to better specify the meaningful results reported also in the tables, which being long could create problems in the identification of the statistically different molecules. Moreover, the use of box plots allows to have a "real" view of the variability of the single metabolite both by age and by sex and highlights better what we wanted to underline in our study.
Round 2
Reviewer 1 Report
I appreciate the effort the authors made to improve the manuscript and to take in consideration the comments and suggestions. With the change on the title the manuscript matches better the expectations. As well as the changes on the text make now the manuscript suitable to be considered for publication in this Journal.
Reviewer 3 Report
Review of manuscript ijms-671291 entitled “Influence of sex on urinary organic acids: a cross-sectional study in children”
I appreciate the efforts done by the authors to response most of the questions raised in my previous review.
However, I feel that two of my previous comments have not been answered and I think that they will improve the quality of the manuscript: